Accepted at the ICLR 2024 Workshop on AI4Differential Equations In Science

# Continuous-time neural networks for modeling linear dynamical systems

**Chinmay Datar**
Institute of Advanced Study
Technical University of Munich, Germany
`chinmay.datar@tum.de`

**Adwait Datar**
Institute for Data-Science Foundations
Technische Universität Hamburg, Germany
`adwait.datar@tuhh.de`

**Felix Dietrich**
School of Computation, Information, and Technology
Technical University of Munich, Germany
`felix.dietrich@tum.de`

**Wil Schilders**
Dept. of Mathematics and Computer Science
Eindhoven University of Technology, Netherlands
`w.h.a.schilders@TUE.nl`

## ABSTRACT

We propose to model Linear Time-Invariant (LTI) systems as a first step towards constructing sparse neural networks for modeling more complex dynamical systems. We use a variant of continuous-time neural networks in which the output of each neuron evolves continuously as a solution of a first or second-order Ordinary Differential Equation (ODE). Instead of computing the network parameters from data, we rely on system identification techniques to obtain a state-space model. Our algorithm is gradient-free, numerically stable, and computes a sparse architecture together with all network parameters from the given state-space matrices of the LTI system. We provide an upper bound on the numerical errors for our constructed neural networks and demonstrate their accuracy by simulating the transient convection-diffusion equation.

## 1 INTRODUCTION

From the evolution of quantum systems to the evolution of celestial bodies, most models in science and engineering are represented as dynamical systems in the form of differential equations. The exploration of neural networks in the modeling of dynamical systems remains an active research field, especially from the perspective of optimization, control, and forecasting Böttcher et al. (2022); Kumpati & Kannan (1990); Linot et al. (2023). There have been considerable strides in modeling sequential and temporal data typically encountered in dynamical systems using discrete-time recurrent neural networks which operate iteratively and discretely on hidden states Kim & Cho (2019); Chimmula & Zhang (2020), and using continuous-time neural networks that model a continuous evolution of hidden states between observations Rubanova et al. (2019); Lechner & Hasani (2020); Gholami et al. (2019). However, many challenges are becoming apparent as well.

There are well-known difficulties in training discrete-time Recurrent Neural Networks (RNNs) using gradient-based approaches such as exploding and vanishing gradients, especially if the data contains temporal dependencies over long intervals Bengio et al. (1994); Mikhaeil et al. (2022); Hochreiter et al. (2001). The difficulties with gradient-based optimization persist for linear Li et al. (2021) and non-linear continuous-time neural networks Meijer (1996). For a special class of dynamical systems, namely Linear Time-Invariant (LTI) systems, Schilders (2009) proposes using a state-space modeling algorithm Verhaegen & Dewilde (1992); Verhaegen (1993) to first identify an LTI system from data, and use it to construct a suitable architecture and compute network parameters gradient-free. Though the system identification algorithm does the heavy lifting in this case, this approach provides insights

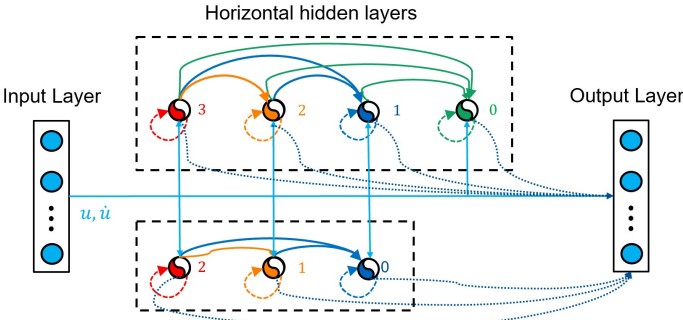

**Figure 1:** Illustrative example of our dynamic neural network with two horizontal hidden layers (top hidden layer with 4 neurons, bottom hidden layer with 3 neurons). The dashed self-connections indicate that the state of each neuron at a future time depends on its state at the previous time. All neurons in the hidden layer have a connection to the input layer. All neurons in the hidden and input layers are connected to the output layer.

into designing sparse and accurate neural networks with appropriate model capacity to model the LTI system. The work by Schilders (2009) is restricted to a class of LTI systems, namely - those with the state matrix having distinct and well-separated eigenvalues. We relax this constraint and propose an algorithm to construct neural networks for arbitrary LTI systems.

Constructing sparse models is a big challenge for complex systems appearing in science and engineering. Several ways of introducing inductive biases in the model design, such as equivariance, invariance, symmetries, and recurrence, have been proposed Karniadakis et al. (2021). However, most of the traditional approaches still involve extensive trial and error and the exploration of numerous architectures, which often entail significant computational costs Elsken et al. (2019). Moreover, the exact model size and capacity required for a task remain unknown, and a common strategy is to train over-parameterized models Li et al. (2020). **In this contribution, we answer the following question: Given a mathematical model—here, an LTI system—can we convert it into a neural network, including its full architecture and parameters?**

We show that the properties of a given LTI system can be used to construct model-based, sparse neural network architectures with a particular topology. Interestingly, the pre-processing transformations we propose and the resulting structure of the state matrix suggest that one should think in terms of horizontal layers for our neural networks (see Section 2). Our key contributions are as follows.

1. We propose Algorithm A.1 to pre-process a given LTI system and Algorithm A.2 to construct a sparse neural network using the properties of the given LTI system.

2. We derive a mapping from the parameters of the LTI system (state-space matrices) to the parameters of the neural network by preserving the input-output map (see Theorem A.1).

3. We give an upper bound on the numerical error of our neural networks (see Theorem A.2).

4. We empirically demonstrate that the neural networks constructed with our proposed algorithm can simulate the LTI system accurately (see Section 3).

A natural question arises at this point: why model LTI systems using neural networks? We emphasize that the goal of this work is not to compete with or replace the existing numerical solvers for simulating LTI systems. The motivation behind choosing LTI systems is to start the mathematical exploration of constructing sparse and accurate neural network models in an easier setting. We view this work as a stepping stone toward constructing appropriate neural network models of more complex dynamical systems using mathematical models or data, or both.

## 2 CONSTRUCTING DYNAMIC NEURAL NETWORKS FROM LTI SYSTEMS

**Dynamic Neural Networks** In this section, we describe a variant of continuous-time neural networks we use, termed "Dynamic Neural Networks" (DyNNs) Meijer (1996). The key difference between classical and dynamic neural networks is that the output of each neuron in the hidden layer of a DyNN is a solution of a first- or second-order ODE. In contrast to the typical neural network architectures, we define a dynamic neural network consisting of "horizontal" layers, in which the

neurons within the same hidden layer have connections as shown in Figure 1. Note that the neurons of a DyNN in different horizontal layers are not connected. The input and output layers of a DyNN are not horizontal.

The neural network architecture describing how neurons are interconnected with each other is shown in Figure 1. Let $n_l$, $d_i$, and $d_o$ be the number of neurons in the horizontal layer $l$, the input layer, and the output layer of a DyNN, respectively. We now define the input-output map of each neuron.

**Definition 1.** *The input-output map of a neuron $i$ in hidden layer $l$ with $d_i^{(l)}$ inputs is a map $f_i^{(l)} : \mathcal{C}(\Omega)^{d_i^{(l)}} \ni u_i^{(l)} \mapsto y_i^{(l)} \in \mathcal{C}^1(\Omega)^2$ defined via the solution to the differential equation*

$$m_i^{(l)} \; \ddot{\xi}_i^{(l)}(t) + c_i^{(l)} \; \dot{\xi}_i^{(l)} + k_i^{(l)} \xi_i^{(l)}(t) = w_i^{(l)} u_i^{(l)}(t), \quad \xi_i^{(l)}(0) = 0, \quad \dot{\xi}_i^{(l)}(0) = 0, \tag{1}$$

$$y_i^{(l)}(t) = \left[ \xi_i^{(l)}(t) \quad \dot{\xi}_i^{(l)}(t) \right]^T \tag{2}$$

*where $w_i^{(l)} \in \mathbb{R}^{1 \times d_i^{(l)}}$, $m_i^{(l)}, c_i^{(l)}, k_i^{(l)} \in \mathbb{R}$ are the weights, and $\xi_i^{(l)}$ is the state of the neuron $i$. The map $f_i^{(l)}$ is defined corresponding to $(m_i^{(l)}, c_i^{(l)}, k_i^{(l)}, w_i^{(l)})$. If $m_i^{(l)} = 0$, we refer to the neuron as a "first-order neuron", otherwise, it is called a "second-order neuron".*

The architecture shown in Figure 1 can be summarized by the functions $f_i$, where

$$y_i^{(l)} = f_i^{(l)}(u_i^{(l)}), \quad \text{where,} \quad u_i^{(l)} = \left[ u^T \quad \dot{u}^T \quad [y_{i+1}^{(l)}]^T \quad [y_{i+2}^{(l)}]^T \quad \cdots \quad [y_{n_l-1}^{(l)}]^T \right]^T, \tag{3}$$

for input coordinate $i \in \{0, \cdots, n_l - 1\}$ and layer $l \in \{1, \cdots, L\}$. The hidden layers of a DyNN represent a coupled system of ODEs whose parameters are $(m_i^{(l)}, c_i^{(l)}, k_i^{(l)}, w_i^{(l)})$. The output layer of a dynamic neural network is a linear layer with connections from all neurons in the hidden and input layers with parameters $\phi_i^{(l)} \in \mathbb{R}^{d_o \times 2}$ and $\Psi \in \mathbb{R}^{d_o \times d_i}$, respectively. The definitions of input-state-output maps and parameter sets (required for theoretical results) are in Appendix A.1. Note that the input-output map of each neuron is essentially an operator described via the solution to an ordinary differential equation. One can interpret each neuron and, thus, the entire network as a response to a continuous function of time. We use the terminology introduced in Meijer (1996) and call our network a 'dynamic' neural network.

**LTI Systems and Pre-processing**  All LTI systems are determined by four matrices: state matrix $A \in \mathbb{R}^{d_h \times d_h}$, input matrix $B \in \mathbb{R}^{d_h \times d_i}$, output matrix $C \in \mathbb{R}^{d_o \times d_h}$ and feed-forward matrix $D \in \mathbb{R}^{d_o \times d_i}$ for $d_h, d_o, d_i \in \mathbb{N}$. The state-space representation of a general LTI system is

$$\dot{x}(t) = A \, x(t) + B \, u(t), \quad x(0) = 0, \tag{4a}$$

$$y(t) = C \, x(t) + D \, u(t), \tag{4b}$$

where $x(t) \in \mathbb{R}^{d_h}$ is the state of the system, $u(t) \in \mathbb{R}^{d_i}$ is the input to the system, and $y(t) \in \mathbb{R}^{d_o}$ is the output of the system. We transform a given LTI system to block-diagonalize the state matrix $A$ in a numerically stable way with the number of blocks $L$ equal to the number of clusters of closely spaced eigenvalues of the state matrix (see Algorithm A.1). The block-diagonalization facilitates the construction of sparser dynamic neural networks. After pre-processing, the new LTI system in the new state coordinates $\xi(t)$ is

$$\dot{\xi}(t) = \tilde{A} \, \xi(t) + \tilde{B} \, u(t), \quad \xi(0) = 0 \tag{5a}$$

$$y(t) = \tilde{C} \, \xi(t) + D \, u(t), \tag{5b}$$

$$\text{where } \tilde{A} = \mathcal{T}^{-1} A \mathcal{T} = \text{blkdiag}[\tilde{A}_{11}, \ldots, \tilde{A}_{LL}], \quad \tilde{B} = \mathcal{T}^{-1} B, \quad \tilde{C} = \mathcal{T} C. \tag{5c}$$

**Mapping from Parameters of the LTI System to Parameters of the DyNN**  We construct a mapping from the parameters of an arbitrary LTI system (state-space matrices) to the parameters of the DyNN (see Theorem A.1). Using this mapping, we compute the parameters and the architecture of the DyNN using the properties of the LTI system in a gradient-free manner. A detailed discussion on this is out of the scope of this paper (see Section A.3). Importantly, the sparsity patterns of the diagonal blocks of the transformed state matrix $\tilde{A}$ in equation 5a (see Section A.2) result in the DyNN architecture consisting of horizontal layers as shown in Table 1 for $\tilde{A}_{ll} \in \mathbb{R}^{4 \times 4}$ and $\tilde{B}^{(l)}$ representing the corresponding four rows of $\tilde{B}$. (see Table A.1 for more illustrations).

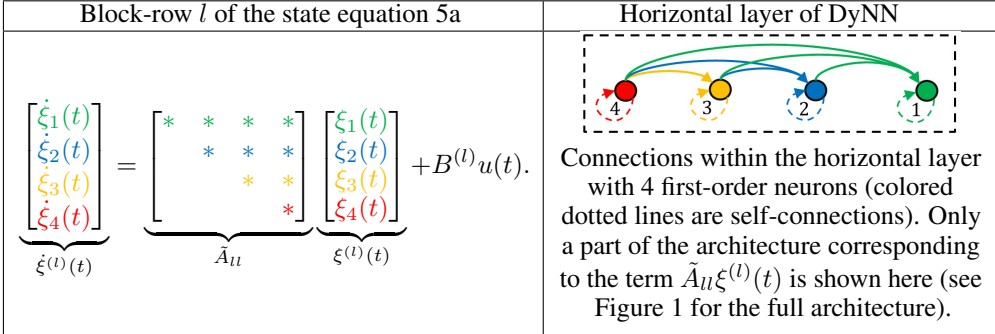

| Block-row $l$ of the state equation 5a | Horizontal layer of DyNN |
|---|---|
| $$\underbrace{\begin{bmatrix} \dot{\xi}_1(t) \\ \dot{\xi}_2(t) \\ \dot{\xi}_3(t) \\ \dot{\xi}_4(t) \end{bmatrix}}_{\dot{\xi}^{(l)}(t)} = \underbrace{\begin{bmatrix} * & * & * & * \\ & * & * & * \\ & & * & * \\ & & & * \end{bmatrix}}_{\tilde{A}_{ll}} \underbrace{\begin{bmatrix} \xi_1(t) \\ \xi_2(t) \\ \xi_3(t) \\ \xi_4(t) \end{bmatrix}}_{\xi^{(l)}(t)} + B^{(l)}u(t).$$ | Connections within the horizontal layer with 4 first-order neurons (colored dotted lines are self-connections). Only a part of the architecture corresponding to the term $\tilde{A}_{ll}\xi^{(l)}(t)$ is shown here (see Figure 1 for the full architecture). |

**Table 1:** Connections between neurons dictated by a block of state-matrix with 4 close eigenvalues

**Algorithm and Numerical Analysis for Dynamic Neural Networks** Our implementation of dynamic neural networks is summarized in Algorithms A.2 and A.3. The code will be made available upon acceptance. We also provide an upper bound on the numerical error of our dynamic neural networks building on the guarantees provided by the ODE solvers (see Theorem A.2). To be precise, we show that if the solvers used in each neuron are known to have an error estimate of $\mathcal{O}(h^p)$, the error estimates of the DyNN are also $\mathcal{O}(h^p)$.

## 3 NUMERICAL EXAMPLES AND DISCUSSION

**Convection-Diffusion Equation:** As a test case, we consider a two-dimensional transient convection-diffusion equation,

$$\frac{\partial T}{\partial t} = \mathcal{D}\Big(\frac{\partial^2 T}{\partial x^2} + \frac{\partial^2 T}{\partial y^2}\Big) - v_x\frac{\partial T}{\partial x} - v_y\frac{\partial T}{\partial y} + \mathcal{S}, \tag{6}$$

where $T$ is the variable of interest (concentration of species or temperature), $\mathcal{D}$ is diffusivity, $v_x$ and $v_y$ are drift velocities in $x$ and $y$ directions, and $S$ is the source term. We interpret this as a system with $S$ as the input and the solution $T$ as the output. The detailed problem setup is discussed in Appendix C.1. We simulate the LTI system obtained by discretizing equation 6 in space using finite differences, with a DyNN and compare the results with ones obtained from the classical numerical solver (Python routine `scipy.signal.lsim`).

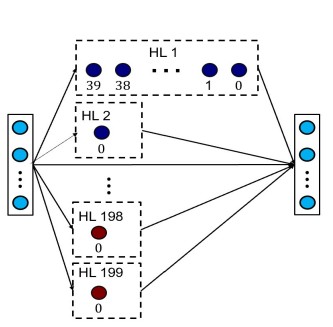
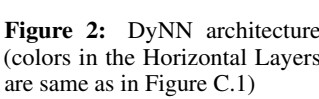
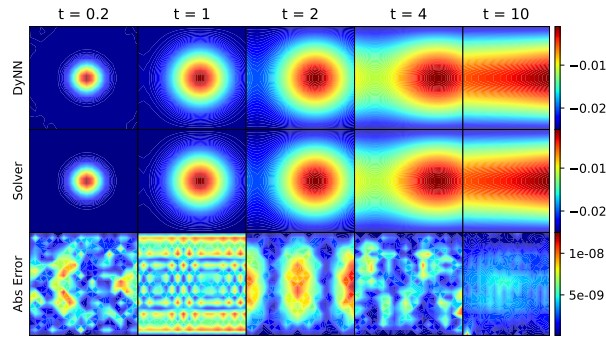

**Figure 2:** DyNN architecture (colors in the Horizontal Layers are same as in Figure C.1)

**Figure 3:** Convection-diffusion equation: dynamic neural network solution (top panel), numerical solution using a python routine scipy.signal.lsim (middle panel), the absolute error between the two solutions (bottom panel) at five-time instances shown at the top.

In this example, the state matrix has a repeated eigenvalue zero, with an algebraic multiplicity of 40 (see Figure C.1 for eigenvalue clustering). For this case, the proposed Algorithms for pre-processing (see Algorithm A.1) and constructing the DyNN (see Algorithm A.2) result in an architecture in which the first horizontal hidden layer has 40 neurons and the rest have one neuron each, as shown in Figure 2. Finally, Figure 3 demonstrates that our constructed DyNN simulates the semi-discretized convection-diffusion system accurately (see Algorithm A.3).

Gradient-free computation of network parameters implies that any black-box and even non-differentiable ODE solver could be used to compute the state of each neuron in the forward pass. Moreover, since the block-diagonalization decouples the slow and fast dynamics across different horizontal layers, the ODE solvers for different neurons can efficiently use an appropriate number of time steps using adaptive time-stepping schemes, and different neurons can even use different ODE solvers (see Figure C.2).

**Future work:** In the future, we intend to extend the idea of a DyNN towards constructing neural networks (including architecture and their parameters) for non-linear dynamical systems. It does not suffice to only apply nonlinear activation functions instead of our current linear maps, because it is unclear in the nonlinear setting how to combine these nonlinear functions to form the target vector field. We will start with simpler, only slightly nonlinear systems in low dimensions to understand the interactions of their vector field and corresponding neural networks. Enforcing known types of dynamics in individual neurons other than the first- and second-order ODEs in DyNN may also help construct larger and non-linear networks.

ACKNOWLEDGMENTS

We would like to acknowledge many helpful discussions from Zahra Monfared, Rahul Manavalan, Iryna Burak, Erik Bolager, Ana Cukarska, Karan Shah, and Qing Sun. While preparing this work, the authors used Grammarly to polish written text for spelling, grammar, and general style.

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

# A  APPENDIX

## A.1  DYNAMIC NEURAL NETWORK: DEFINITIONS

**Definition 2** (Parameter sets of the dynamic neural network). *For positive integers $d_i$, $d_o$, $L$ and $n_l, l \in \{1, \cdots L\}$, let the tuple of weights $m_i^{(l)}$ in layer $l$ be*

$$\mathcal{M}^{(l)} = \left( m_0^{(l)}, \cdots, m_{n_l-1}^{(l)} \right) \quad \text{and let} \quad \mathcal{M} = \left( \mathcal{M}^{(1)}, \cdots, \mathcal{M}^{(L)} \right).$$

*Similarly define $\mathcal{C}^{(l)}, \mathcal{C}$ from all $c_i^{(l)}$; $\mathcal{K}^{(l)}, \mathcal{K}$ from all $k_i^{(l)}$; $\mathcal{W}^{(l)}, \mathcal{W}$ from all $w_i^{(l)}$ and $\Phi^{(l)}, \Phi$ from all $\phi_i^{(l)}$. Finally, define the sets*

$$\mathcal{P}_{dynn}^{(l)} := \left\{ \left( \mathcal{M}^{(l)}, \mathcal{C}^{(l)}, \mathcal{K}^{(l)}, \mathcal{W}^{(l)} \right) : m_i^{(l)}, c_i^{(l)}, k_i^{(l)} \in \mathbb{R}, w_i^{(l)} \in \mathbb{R}^{1 \times d_i^{(l)}} \right\},$$

$$\mathcal{P}_{dynn}^{hidden} := \left\{ (\mathcal{M}, \mathcal{C}, \mathcal{K}, \mathcal{W}) : m_i^{(l)}, c_i^{(l)}, k_i^{(l)}, w_i^{(l)} \in \mathbb{R}^{1 \times d_i^{(l)}} \right\},$$

$$\mathcal{P}_{dynn}^{output} := \left\{ (\Phi, \Psi) : \phi_i^{(l)} \in \mathbb{R}^{d_o \times 2}, \Psi \in \mathbb{R}^{d_o \times d_i} \right\}$$

*which collect all parameters of the hidden layer $l$, all parameters of all hidden layers together and all parameters of the output layer, respectively.*

**Definition 3** (Input-state-output maps of DyNN). *Consider a dynamic neural network with $L$ horizontal layers, $n_l$ neurons in the horizontal layer $l$, $d_i$ neurons in the input layer and $d_o$ neurons in the output layer. Let $(\mathcal{M}, \mathcal{C}, \mathcal{K}, \mathcal{W}) \in \mathcal{P}_{dynn}^{hidden}$ and $(\Phi, \Psi) \in \mathcal{P}_{dynn}^{output}$ be the parameters of the DyNN. The forward pass of the DyNN for an arbitrary input $u \in \mathcal{C}^1(\Omega)^{d_i}$ can be described via the equations*

$$y(t) = \left( \sum_{l=1}^{L} \sum_{i=0}^{n_l-1} \phi_i^{(l)} y_i^{(l)}(t) \right) + \Psi u(t), \quad \text{for } t \in \Omega, \tag{7}$$

$$y_i^{(l)}(t) = \begin{bmatrix} \xi_i^{(l)}(t) \\ \dot{\xi}_i^{(l)}(t) \end{bmatrix} = f_i^{(l)}(u_i^{(l)})(t), \tag{8}$$

$$u_i^{(l)}(t) = \begin{bmatrix} u^T(t) & \dot{u}^T(t) & [y_{i+1}^{(l)}(t)]^T & [y_{i+2}^{(l)}(t)]^T & \cdots & [y_{n_l-1}^{(l)}(t)]^T \end{bmatrix}^T, \tag{9}$$

*where $f_i^{(l)}$ is the input-output map corresponding to $(m_i^{(l)}, c_i^{(l)}, k_i^{(l)}, w_i^{(l)})$ described in Definition 1. Based on these equations, the **input-output map of DyNN** $f_{dynn} : \mathcal{C}^1(\Omega)^{d_i} \to \mathcal{C}^1(\Omega)^{d_o}$, the **input-state map of DyNN** $f_{dynn}^s : \mathcal{C}^1(\Omega)^{d_i} \to \mathcal{C}^1(\Omega)^{2(n_1+\cdots+n_L)}$ and the **input-state map of the** $l^{\text{th}}$ **hidden layer of DyNN** $f_{dynn}^{(l)} : \mathcal{C}^1(\Omega)^{d_i} \to \mathcal{C}^1(\Omega)^{2n_l}$ are defined as*

$$f_{dynn} : u \mapsto y,$$

$$f_{dynn}^s : u \mapsto \xi, \text{ where } \xi(t) = \begin{bmatrix} \xi^{(1)}(t) \\ \vdots \\ \xi^{(L)}(t) \end{bmatrix},$$

$$f_{dynn}^{(l)} : u \mapsto \xi^{(l)}, \text{ where } \xi^{(l)}(t) = \begin{bmatrix} \xi_0^{(l)}(t) \\ \vdots \\ \xi_{n_l-1}^{(l)}(t) \end{bmatrix}, \quad l \in \{1, \cdots, L\},$$

$$\dot{f}_{dynn}^{(l)} : u \mapsto \dot{\xi}^{(l)}, \text{ where } \dot{\xi}^{(l)}(t) = \begin{bmatrix} \dot{\xi}_0^{(l)}(t) \\ \vdots \\ \dot{\xi}_{n_l-1}^{(l)}(t) \end{bmatrix}, \quad l \in \{1, \cdots, L\}.$$

*We call $\xi^{(l)}$ the state of the horizontal layer $l$, and $\xi$ is called the state of the DyNN.*

---

**Algorithm A.1** Pre-processing the LTI system

---

**Input:** State Space Matrices($A, B, C, D$)
**Output:** Transformed State Space Matrices ($\tilde{A}, \tilde{B}, \tilde{C}, \tilde{D}$)
**Parameters:** `Clustering algorithm`

  1: $\mathcal{R} \leftarrow \mathcal{T}_1^T A \mathcal{T}_1$               // Real Schur decomposition
  2: **if** $\mathcal{R}$ is diagonal **then**       // If A is unitarily diagonalizable
  3:      $\mathcal{T} = \mathcal{T}_1$             // Transformation matrix
  4:      $\tilde{A} \leftarrow \mathcal{R}$
  5: **else if** $\mathcal{R}$ is not diagonal **then**    // If A is not unitarily diagonalizable
  6:      $\tilde{\mathcal{R}} \leftarrow \mathcal{T}_2^T A \mathcal{T}_2$         // Ordered real Schur form (Bai & Demmel (1993))
  7:      $\tilde{A} \leftarrow \mathcal{T}_3^{-1} \tilde{\mathcal{R}} \mathcal{T}_3$       // Block-diagonalization (Bartels & Stewart (1972))
  8:      $\mathcal{T} = \mathcal{T}_2 \mathcal{T}_3$          // Total transformation matrix
  9: **end if**
10: $(\tilde{B}, \tilde{C}, \tilde{D}) \leftarrow (\mathcal{T}^{-1} B, C \mathcal{T}, D)$   // New State Space Matrices

---

## A.2 PRE-PROCESSING ALGORITHM

In the first step of Algorithm A.1, we perform the real Schur decomposition $\mathcal{R} = \mathcal{T}_1^T A \mathcal{T}_1$, where $\mathcal{T}_1$ is orthogonal and the diagonal blocks are $\mathbb{R}^{1 \times 1}$ for real eigenvalues or $\mathbb{R}^{2 \times 2}$ for complex pairs of eigenvalues. If the matrix $A$ is unitarily diagonalizable, $\mathcal{R}$ becomes a diagonal matrix. This is the ideal case when one can unitarily diagonalize the state matrix in a numerically stable way.

However, if this is not the case, we proceed with a modified version of the ordered real Schur decomposition proposed in Bai & Demmel (1993). Specifically, we implement an algorithm that specifies the order in which the eigenvalues appear on the diagonal such that the eigenvalues that are close to each other can be re-grouped as bigger diagonal blocks. Thus, the transformation $\tilde{\mathcal{R}} \leftarrow \mathcal{T}_2^T A \mathcal{T}_2$ reduces the state matrix to a block-upper triangular matrix such that the eigenvalues in different diagonal blocks are well-separated. The clustering of close eigenvalues in respective diagonal blocks and ensuring that eigenvalues in different diagonal blocks are well-separated is necessary to apply the Bartels-Stewart algorithm in a numerically stable way. The Bartels-Stewart algorithm is a similarity transformation $\mathcal{T}_3^{-1} \tilde{\mathcal{R}} \mathcal{T}_3$ that can reduce all the off-diagonal entries of $\tilde{\mathcal{R}}$ to zero, and block-diagonalize the state matrix.

The parameter `Clustering algorithm` in Algorithm A.1 can be chosen as any of the vast variety of clustering algorithms to cluster eigenvalues of the state matrix $A$. These include the well-known k-means algorithm Lloyd (1982), the spectral clustering algorithm Von Luxburg (2007), and others. These algorithms and many more are implemented in the Python package scikit learn Pedregosa et al. (2011). For each cluster of eigenvalues, we identify the eigenvalue with the largest real part and sort the clusters in descending order based on these. This step is not necessary, but it ensures that the algorithm is deterministic. Within each cluster of eigenvalues, we order the eigenvalues according to the absolute value of the real part in ascending order. This is required to ensure that for a cluster having real and complex eigenvalues, the real eigenvalues are placed first, which is exploited in Theorem A.1.

After block-diagonalization of the state-matrix, each diagonal block has either all real, all complex, or mixed eigenvalues that are close to each other. We define the set of state matrices with this sparsity pattern in the following definition.

**Definition 4** (Sets of sparse state matrices). *Let $\mathcal{G}$ be the set of all block-upper triangular matrices $M$ such that*

    *1. $M$ has $k_r$ real eigenvalues for some $k_r \in \mathbb{N}$,*

    *2. $M$ has $k_c$ pairs of complex eigenvalues with non-zero imaginary parts for some $k_c \in \mathbb{N}$,*

    *3. all blocks in the first $k_r$ rows of $M$ are of dimension $1 \times 1$ and all blocks in the last $2k_c$ rows of $M$ are of dimension $2 \times 2$,*

    *4. All $2 \times 2$ blocks on the diagonal in the last $2k_c$ rows of $M$ have non-zero entries in the upper-right corner, i.e., if $\begin{bmatrix} a & b \\ c & d \end{bmatrix}$ is a diagonal block in the last $2k_c$ rows, then $b \neq 0$.*

*Let $\mathcal{G}_r \subset \mathcal{G}$ be the subset containing matrices having all real eigenvalues, i.e., $k_c = 0$ and let $\mathcal{G}_c \subset \mathcal{G}$ be the subset containing matrices with all eigenvalues having non-zero imaginary parts, i.e., $k_r = 0$.*

**Definition 5** (Parameters of an LTI system). *Corresponding to positive integers $d_h, d_i, d_o$, define*

$$\mathcal{P}_{lti}^{state} := \left\{ (A, B) : A \in \mathcal{S} \subset \mathbb{R}^{d_h \times d_h}, B \in \mathbb{R}^{d_h \times d_i} \right\},$$
$$\mathcal{P}_{lti}^{output} := \left\{ (C, D) : C \in \mathbb{R}^{d_o \times d_h}, D \in \mathbb{R}^{d_o \times d_i} \right\},$$

*where $\mathcal{S}$ is the set of all square matrices that are block-diagonal, with each diagonal block belonging to $\mathcal{G}$.*

**Definition 6** (Input-state-output maps of an LTI system). *Let $A \in \mathbb{R}^{d_h \times d_h}, B \in \mathbb{R}^{d_h \times d_i}, C \in \mathbb{R}^{d_o \times d_h}, D \in \mathbb{R}^{d_o \times d_i}$. Let the state $x \in \mathcal{C}^1(\Omega)^{d_h}$, input $u \in \mathcal{C}(\Omega)^{d_i}$ and output $y \in \mathcal{C}(\Omega)^{d_o}$ be related by the governing equations of an LTI system*

$$\dot{x}(t) = Ax(t) + Bu(t), \quad x(0) = 0, \tag{10}$$
$$y(t) = Cx(t) + Du(t), \tag{11}$$

*for $t \in \Omega$. For this LTI system, define the **input-state map of the LTI system** corresponding to $(A, B)$ as $f_{lti}^s : \mathcal{C}(\Omega)^{d_i} \ni u \mapsto x \in \mathcal{C}^1(\Omega)^{d_h}$ defined via equation 10 and the **input-output map of the LTI system** corresponding to $(A, B, C, D)$ as $f_{lti} : \mathcal{C}(\Omega)^{d_i} \ni u \mapsto y \in \mathcal{C}(\Omega)^{d_o}$ defined via equation 10 and equation 11.*

The sparsity patterns of the diagonal blocks, depending on whether they have real only, complex only, or mixed eigenvalues, are

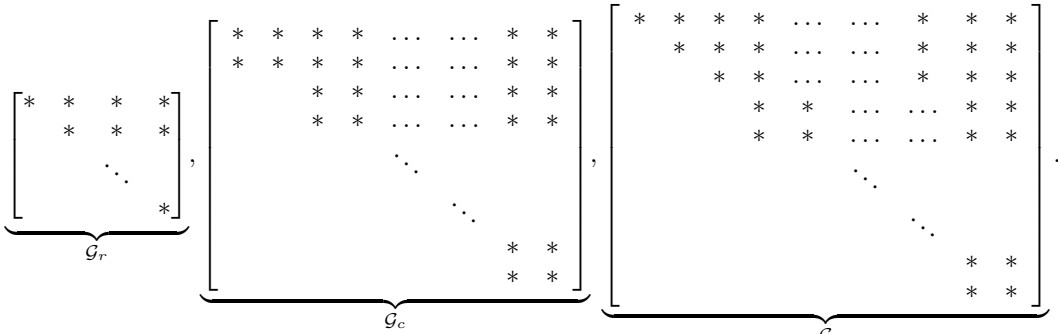

### A.3 Computing DyNN architecture and parameters from parameters of the LTI System

**Summary of this section:** This section builds on the definitions in sections A.1 and A.2. We first note that the input-output map of our DyNN can be described as the solution of a coupled system of second-order differential equations. This can be seen by assembling the second-order differential equations corresponding to all neurons and substituting the interconnection structure (see Lemma A.1). As an intermediate technical result, we show that the input-output map of an LTI system can also be represented as the solution of a system of second-order differential equations (see Lemma A.2). Finally, we derive a mapping from the parameters of the LTI system to the parameters of the DyNN by preserving the input-output map (see Theorem A.1).

We start by defining the set of tuples of matrices that describe a second-order coupled system of ODEs, which is represented by each horizontal layer of the DyNN.

**Definition 7** (Tuples of matrices defining second order system). *Corresponding to $n_l, d_i \in \mathbb{N}$, let $\mathcal{S}_{n_l, d_i}$ be the set of tuples $(M, C, K, E, V)$ where $M, C, K \in \mathbb{R}^{n_l \times n_l}, E, V, \in \mathbb{R}^{n_l \times d_i}$, $M$ is a diagonal matrix and $C$ and $K$ are upper-triangular matrices.*

**Lemma A.1** (First and/or Second order dynamics of DyNN). *Consider a dynamic neural network with $L$ horizontal layers, $n_l$ neurons in the horizontal layer $l$, $d_i$ neurons in the input layer, and $d_o$ neurons in the output layer. For $l \in \{1, \cdots, L\}$, let $\left(\mathcal{M}^{(l)}, \mathcal{C}^{(l)}, \mathcal{K}^{(l)}, \mathcal{W}^{(l)}\right) \in \mathcal{P}_{dynn}^{(l)}$ be the parameters of the hidden layers of the DyNN. Let $u \in \mathcal{C}^1(\Omega)^{d_i}$ be an arbitrary input and $\xi^{(l)}$ be the state of the $l^{\text{th}}$ hidden layer, i.e., $\xi^{(l)} = f_{dynn}^{(l)}(u)$. Then for $l \in \{1, \cdots, L\}$, a bijective mapping*

$$\mathfrak{n}_{dynn}^{(l)} : \mathcal{P}_{dynn}^{(l)} \ni \left(\mathcal{M}^{(l)}, \mathcal{C}^{(l)}, \mathcal{K}^{(l)}, \mathcal{W}^{(l)}\right) \mapsto \left(M^{(l)}, C^{(l)}, K^{(l)}, E^{(l)}, V^{(l)}\right) \in \mathcal{S}_{n_l, d_i},$$

*described in Appendix B.2 can be constructed such that $\xi^{(l)}$ satisfies $\xi^{(l)}(0) = 0$, $\dot{\xi}^{(l)} = 0$ and*

$$M^{(l)}\ddot{\xi}^{(l)}(t) + C^{(l)}\dot{\xi}^{(l)}(t) + K^{(l)}\xi^{(l)}(t) = E^{(l)}u(t) + V^{(1)}\dot{u}(t) \quad \forall t \in \Omega. \tag{12}$$

*Conversely, for $l \in \{1, \cdots, L\}$ and arbitrary $\left(M^{(l)}, C^{(l)}, K^{(l)}, E^{(l)}, V^{(l)}\right) \in \mathcal{S}_{n_l, d_i}$, if $\xi^{(l)}$ solves the differential equation equation 12 with zero initial conditions, then one can construct a DyNN with parameters $\left(\mathcal{M}^{(l)}, \mathcal{C}^{(l)}, \mathcal{K}^{(l)}, \mathcal{W}^{(l)}\right)$ computed by the inverse of $\mathfrak{n}_{dynn}^{(l)}$ such that $\xi^{(l)} = f_{dynn}^{(l)}(u)$ and $\dot{\xi}^{(l)} = \dot{f}_{dynn}^{(l)}(u)$.*

**Lemma A.2** (Second order dynamics from an LTI system). *Let $\mathcal{A} \in \mathcal{G} \subset \mathbb{R}^{(k_r + 2k_c) \times (k_r + 2k_c)}$ and $\mathcal{B} \in \mathbb{R}^{(k_r + 2k_c) \times d_i}$ for some non-negative integers $k_r, k_c$ and $d_i \in \mathbb{N}$. Let the input $u \in \mathcal{C}^1(\Omega)^{d_i}$ and state $x \in \mathcal{C}^2(\Omega)^{(k_r + 2k_c)}$ satisfy the linear differential equation*

$$\dot{x} = \mathcal{A}x + \mathcal{B}u, \quad x(0) = 0.$$

*The mappings*

$$\mathfrak{m}_{lti} : (\mathcal{A}, \mathcal{B}) \mapsto (M, C, K, E, V) \in \mathcal{S}_{k_r + k_c} \quad \text{(see Definition 7)},$$
$$\mathfrak{m}_{\eta} : (\mathcal{A}, \mathcal{B}) \mapsto (W, Q, Z)$$

*as described in Appendix B.1 can be constructed such that the new variables $\xi(t) \in \mathbb{R}^{k_r + k_c}$, $\eta(t) \in \mathbb{R}^{k_c}$, $\xi_r(t) \in \mathbb{R}^{k_r}$ and $\xi_c(t) \in \mathbb{R}^{k_c}$ defined as*

$$\left[\frac{\xi(t)}{\eta(t)}\right] := \left[\begin{array}{c} \xi_r(t) \\ \xi_c(t) \\ \hline \eta(t) \end{array}\right] = \left[\begin{array}{cc} I_{k_r} & 0 \\ 0 & I_{k_c} \otimes \begin{bmatrix} 1 & 0 \end{bmatrix} \\ 0 & I_{k_c} \otimes \begin{bmatrix} 0 & 1 \end{bmatrix} \end{array}\right] x(t)$$

*satisfy*

$$M\ddot{\xi}(t) + C\dot{\xi}(t) + K\xi(t) = Eu(t) + V\dot{u}(t), \tag{13}$$
$$\eta(t) = W\,\xi_c(t) + Q\dot{\xi}_c(t) + Zu(t), \tag{14}$$

*for all $t \in \Omega$ with $\xi(0) = 0$, $\eta(0) = 0$. Furthermore, the matrices $W, Q \in \mathbb{R}^{k_c \times k_c}$ are upper-triangular, i.e., $W_{ij} = 0, Q_{ij} = 0$ for $i > j$ and $Z \in \mathbb{R}^{k_c \times d_i}$.*

**Remark 1.** *Depending on whether all, few, or none of the entries of $\mathcal{M}^{(l)}$ are zero, the states of the horizontal layer $l$ of the DyNN form a linear, coupled system of either only first-order ODEs or a combination of first-order and second-order ODEs or only second-order ODEs, respectively.*

**Theorem A.1** (Mapping an LTI system to a DyNN). *For positive constants $d_h, d_i, d_o$, consider an LTI system defined by $(A, B) \in \mathcal{P}_{lti}^{state}$ and $(C, D) \in \mathcal{P}_{lti}^{output}$. For positive integers $L$ and $n_l$ for $l \in \{1, \cdots L\}$, mappings*

$$\mathfrak{m}_h : (A, B) \mapsto (\mathcal{M}, \mathcal{C}, \mathcal{K}, \mathcal{W}, \Theta) \in \mathcal{P}_{dynn}^{hidden},$$
$$\mathfrak{m}_o : (A, B, C, D) \mapsto (\Phi, \Psi) \in \mathcal{P}_{dynn}^{output},$$

*as described in Appendix B.3 and B.4 can be constructed such that the DyNN with parameters $(\mathcal{M}, \mathcal{C}, \mathcal{K}, \mathcal{W}, \Theta)$ and $(\Phi, \Psi)$ satisfies the property that $f_{dynn}(u) = f_{lti}(u)$ for all $u \in \mathcal{C}^1(\Omega)^{d_i}$.*

**Table A.1:** Types of horizontal hidden layers based on the number of real and complex eigenvalues $k_r^{(l)}, k_c^{(l)}$ of a block of a state matrix $\tilde{A}_{ll}$. Note how the connections between neurons are dictated by the sparsity pattern of the individual diagonal blocks of the transformed-state matrix. Colored dotted lines are self-connections. Only a part of the architecture corresponding to the term $\tilde{A}_{ll}\xi^{(l)}(t)$ is shown here (see Figure 1 for the full architecture)

| Eigen-values | State equation of the LTI system | Horizontal layer of DyNN |
|---|---|---|
| $k_r^{(l)} = 4,$ $k_c^{(l)} = 0$ | $\begin{bmatrix} \dot{\xi}_1(t) \\ \dot{\xi}_2(t) \\ \dot{\xi}_3(t) \\ \dot{\xi}_4(t) \end{bmatrix} = \underbrace{\begin{bmatrix} * & * & * & * \\ & * & * & * \\ & & * & * \\ & & & * \end{bmatrix}}_{\tilde{A}_{ll} \in \mathcal{G}_r} \begin{bmatrix} \xi_1(t) \\ \xi_2(t) \\ \xi_3(t) \\ \xi_4(t) \end{bmatrix} + \tilde{B}^{(l)}u(t).$ |  DyNN horizontal layer with four first-order neurons (solid balls) |
| $k_r^{(l)} = 0,$ $k_c^{(l)} = 4$ | $\begin{bmatrix} \dot{\xi}_1(t) \\ \dot{\xi}_2(t) \\ \dot{\xi}_3(t) \\ \dot{\xi}_4(t) \\ \dot{\xi}_5(t) \\ \dot{\xi}_6(t) \\ \dot{\xi}_7(t) \\ \dot{\xi}_8(t) \end{bmatrix} = \underbrace{\begin{bmatrix} * & * & * & * & * & * & * & * \\ * & * & * & * & * & * & * & * \\ & & * & * & * & * & * & * \\ & & * & * & * & * & * & * \\ & & & & * & * & * & * \\ & & & & * & * & * & * \\ & & & & & & * & * \\ & & & & & & * & * \end{bmatrix}}_{\tilde{A}_{ll} \in \mathcal{G}_c} \begin{bmatrix} \xi_1(t) \\ \xi_2(t) \\ \xi_3(t) \\ \xi_4(t) \\ \xi_5(t) \\ \xi_6(t) \\ \xi_7(t) \\ \xi_8(t) \end{bmatrix}$ $+ \tilde{B}^{(l)}u.$ |  DyNN horizontal layer with four second-order neurons (yin-yang balls) |
| $k_r^{(l)} = 2,$ $k_c^{(l)} = 2$ | $\begin{bmatrix} \dot{\xi}_1(t) \\ \dot{\xi}_2(t) \\ \dot{\xi}_3(t) \\ \dot{\xi}_4(t) \\ \dot{\xi}_5(t) \\ \dot{\xi}_6(t) \end{bmatrix} = \underbrace{\begin{bmatrix} * & * & * & * & * & * \\ & * & * & * & * & * \\ & & * & * & * & * \\ & & * & * & * & * \\ & & & & * & * \\ & & & & * & * \end{bmatrix}}_{\tilde{A}_{ll} \in \mathcal{G}} \begin{bmatrix} \xi_1(t) \\ \xi_2(t) \\ \xi_3(t) \\ \xi_4(t) \\ \xi_5(t) \\ \xi_6(t) \end{bmatrix}$ $+ \tilde{B}^{(l)}u(t).$ |  DyNN horizontal layer with two first-order neurons (green and blue solid balls) and two second-order neurons (red and yellow yin-yang balls) |

## A.4  DYNAMIC NEURAL NETWORK ALGORITHM AND NUMERICAL ANALYSIS

Algorithm A.2 takes a state-space model $(A, B, C, D)$ as input and for a selected clustering algorithm, constructs a dynamic neural network architecture and parameters. The input is pre-processed as described in Algorithm A.1. For each diagonal block, based on the number of real and complex eigenvalues of the transformed state matrix $\tilde{A}$, we construct a corresponding horizontal layer with appropriate first and second-order neurons as shown in an illustrative example in Table A.1. Finally, all parameters of horizontal layers and the output layer of the DyNN are computed using the maps $\mathfrak{m}_h$ and $\mathfrak{m}_o$ as described in Appendix B.

Algorithm A.3 takes as input a DyNN with fixed architecture and parameters (output of Algorithm A.2) and inputs to the network $u(t), \dot{u}(t)$ to compute the output of the DyNN. The initial conditions of the ODE to be solved for the state of each neuron are set to zero by default and could also be set to any other value. The properties of the ODE of each neuron, such as the ODE solver denoted by `method`, relative and absolute tolerances `rtol`, `atol` respectively that control the accuracy of the solution and a parameter `dense_output`, are set as defined by the user and can be different for different neurons. The parameter `dense_output` of the method `solve_ivp` is set to true, which means that the output of the ODE is a function handle that can be evaluated by interpolation at any time point $t \in \Omega$. The order of interpolation depends on the method specified. For instance, for RK23, a cubic Hermite polynomial is used. For DOPRI85, a seventh-order polynomial is used. Most of the standard explicit and implicit solvers are implemented in the `solve_ivp` routine of the

SciPy package Virtanen et al. (2020). To name a few - explicit methods such as RK45 Dormand & Prince (1980), RK23 Shampine (1986), and DOPRI85 Wanner & Hairer (1996), as well as implicit methods such as Radau Hairer et al. (1991), BDF Shampine & Reichelt (1997), and LSODA Petzold (1983), are implemented. The output of the DyNN is then computed again as a function handle. Note that line 11 in the algorithm A.3 concerning the output $\hat{y}$ is a functional assignment. The user can easily specify the time points at which the output of the DyNN is to be evaluated.

---

**Algorithm A.2** Computing dynamic neural network architecture and parameters

---

**Input:** State Space Model $(A, B, C, D)$
**Output:** DyNN architecture and parameters - $(\mathcal{M}, \mathcal{C}, \mathcal{K}, \mathcal{W}, \Theta) \in \mathcal{P}_{dynn}^{hidden}, (\Phi, \Psi) \in \mathcal{P}_{dynn}^{output}$
**Parameters:** `Clustering algorithm`

1: **Pre-process the LTI system**
2:     Pre-process the LTI system: $(\tilde{A}, \tilde{B}, \tilde{C}, \tilde{D}) \leftarrow (A, B, C, D)$          // Algorithm A.1
3: **Construct horizontal layers of DyNN**
4: **for** $l \leftarrow 1$ to $L$ **do**
5:     Construct horizontal layer with $k_r^{(l)}$ first-order, $k_c^{(l)}$ second-order neurons     // Table A.1
6:     Compute horizontal layer parameters $\left(\mathcal{M}^{(l)}, \mathcal{C}^{(l)}, \mathcal{K}^{(l)}, \mathcal{W}^{(l)}\right) \in \mathcal{P}_{dynn}^{(l)}$     // Theorem A.1
7: **end for**
8: **Construct output layer of DyNN**
9: Compute output layer parameters $(\Phi, \Psi) \in \mathcal{P}_{dynn}^{output}$          // Theorem A.1

---

**Algorithm A.3** Forward pass of a dynamic neural network

---

**Input:** DyNN architecture and parameters - $(\mathcal{M}, \mathcal{C}, \mathcal{K}, \mathcal{W}, \Theta) \in \mathcal{P}_{dynn}^{hidden}, (\Phi, \Psi) \in \mathcal{P}_{dynn}^{output}$,
inputs $u$ and $\dot{u}$ as function handles
**Output:** Output of the dynamic neural network $\hat{y}$ as a function handle
**Parameters:** `rtol, atol, method`

1: **for** $l \leftarrow 1$ to $L$ **do**
2:     **for** $i \leftarrow n_l - 1$ to $0$ **do**
3:        Set initial conditions $\hat{y}_i^{(l)}(0)$ to 0.
4:        `properties` $\leftarrow$ `method, rtol, atol, dense_output`
5:        `weights` $\leftarrow \left( m_i^{(l)}, c_i^{(l)}, k_i^{(l)}, w_i^{(l)}, \phi_i^{(l)} \right)$
6:        $\hat{u}_i^{(l)} \leftarrow \begin{bmatrix} u^T & \dot{u}^T & \hat{y}_{i+1}^{(l)T} & \hat{y}_{i+2}^{(l)T} & \cdots & \hat{y}_{n_l-1}^{(l)T} \end{bmatrix}^T$
7:        $\hat{y}_i^{(l)} \leftarrow$ `solve_ivp(`$\hat{y}_i^{(l)}(0), \hat{u}_i^{(l)}$`,weights, properties)`
8:     **end for**
9: **end for**
10: Set the remaining output layer weights - $\Psi$
11: Compute DyNN output $\hat{y} \leftarrow \left( \sum_{l=1}^{L} \sum_{i=0}^{n_l-1} \phi_i^{(l)} \hat{y}_i^{(l)} \right) + \Psi u$

---

**Remark 2.** *If the input to the dynamic neural network $u$ is available only at a finite number of time points, then the user can specify how to interpolate $u$. Currently, we provide an option to approximate $u$ with either a piecewise constant function or a piecewise linear function.*

In analogy with input-output maps defined on analytical solutions of the ODEs, we now define the input-output maps for a neuron and a DyNN based on numerical solutions of the ODEs. These maps are then used for the error analysis presented in the next subsection.

**Definition 8** (Input-output map of a numerically implemented neuron)**.** *The **input-output map of the numerically implemented neuron** $i$ in hidden layer $l$ with $d_i^{(l)}$ inputs is a map $\hat{f}_i^{(l)} : \mathcal{C}^1(\Omega)^{d_i^{(l)}} \to \mathcal{C}^2(\Omega)^2$ defined as $\hat{u}_i^{(l)} \mapsto \hat{y}_i^{(l)}$, where $\hat{y}_i^{(l)}$ is the output of the function* `solve_ivp` *used in Algorithm A.3 corresponding to input $\hat{u}_i^{(l)}$ and parameters $(m_i^{(l)}, c_i^{(l)}, k_i^{(l)}, w_i^{(l)})$.*

**Definition 9** (Input-output map of a numerically implemented DyNN)**.** *Corresponding to a given DyNN and the parameters of Algorithm A.3, the **input-output map of a numerically implemented***

*dynamic neural network* with $L$ horizontal layers, $n_l$ neurons in the horizontal layer $l$, $d_i$ neurons in the input layer and $d_o$ neurons in the output layer is defined as $\hat{f}_{dynn} : u \mapsto \hat{y}$ where $\hat{y}$ is the output of Algorithm A.3 corresponding to inputs $(\mathcal{M}, \mathcal{C}, \mathcal{K}, \mathcal{W}, \Theta) \in \mathcal{P}_{dynn}^{hidden}, (\Phi, \Psi) \in \mathcal{P}_{dynn}^{output}$ and inputs $u$ and $\dot{u}$.

**Theorem A.2.** *Assume that function* `solve_ivp` *implemented on all neurons $i \in \{1, \cdots, n_l\}$ in all layers $l \in \{1, \cdots, L\}$ from Algorithm A.3 mapping the input $\hat{u}_i^{(l)}$ to the solution $\hat{y}_i^{(l)}$ satisfies the error bound*

$$||\hat{y}_i^{(l)}(t) - f_i^{(l)}(\hat{u})(t)|| = \mathcal{O}(h^p) \quad \forall t \in \Omega, \forall \hat{u} \in [\mathcal{C}^1(\Omega)]^{d_i^{(l)}}, \tag{15}$$

*where $f_i^{(l)}$ is the input-output map corresponding to neuron $i$ in layer $l$ (see Definition 1). Then we have that*

$$||f_{dynn}(u)(t) - \hat{f}_{dynn}(u)(t)|| = \mathcal{O}(h^p) \quad \forall t \in \Omega, \forall u \in [\mathcal{C}^1(\Omega)]^{d_i},$$

*where $f_{dynn}$ is the input-output map of the dynamic neural network (see Definition 3) and $\hat{f}_{dynn}$ is the input-output map of the numerical implementation of the dynamic neural network (see definition 9), both corresponding to the same parameters.*

# B  MAPPINGS

## B.1  MAPPINGS $\mathfrak{m}_\eta$ AND $\mathfrak{m}_{lti}$:

Assume that $k_r, k_c, d_i, \mathcal{A} \in \mathcal{G} \subset \mathbb{R}^{(k_r+2k_c)\times(k_r+2k_c)}$ and $\mathcal{B} \in \mathbb{R}^{(k_r+2k_c)\times d_i}$ are given. We will next describe the mappings

$$\mathfrak{m}_\eta : (\mathcal{A}, \mathcal{B}) \mapsto (W, Q, Z),$$
$$\mathfrak{m}_{lti} : (\mathcal{A}, \mathcal{B}) \mapsto (M, C, K, E, V) \in \mathcal{S}_{k_r+k_c} \quad \text{(see Definition 7)}.$$

We start by partitioning the matrix $\mathcal{A}$ as

$$\mathcal{A} = \begin{bmatrix} \mathcal{A}_r & \mathcal{A}_{rc} \\ 0 & \mathcal{A}_c \end{bmatrix} \tag{16a}$$

with $\mathcal{A}_r \in \mathcal{G}_r \subset \mathbb{R}^{k_r \times k_r}, \mathcal{A}_c \in \mathcal{G}_c \subset \mathbb{R}^{2k_c \times 2k_c}$ (see Definition 4) and define blocks $\mathcal{A}_{ij}$ and $\mathcal{B}_i$ for $i, j \in \{1, 2, 3\}$ and as

$$\left[\begin{array}{c|cc} \mathcal{A}_{11} & \mathcal{A}_{12} & \mathcal{A}_{13} \\ \hline 0 & \mathcal{A}_{22} & \mathcal{A}_{23} \\ 0 & \mathcal{A}_{32} & \mathcal{A}_{33} \end{array}\right] = \left[\begin{array}{c|c} \mathcal{A}_r & \mathcal{A}_{rc}[T]^T \\ \hline 0 & T\mathcal{A}_c[T]^T \end{array}\right], \quad \left[\begin{array}{c} \mathcal{B}_1 \\ \hline \mathcal{B}_2 \\ \mathcal{B}_3 \end{array}\right] = P\mathcal{B}, \tag{16b}$$

where the blocks $\mathcal{A}_{22}, \mathcal{A}_{23}, \mathcal{A}_{32}, \mathcal{A}_{33} \in \mathbb{R}^{k_c \times k_c}$ and

$$T = \begin{bmatrix} I_{k_c} \otimes [1 \quad 0] \\ I_{k_c} \otimes [0 \quad 1] \end{bmatrix}, \quad P = \begin{bmatrix} I_{k_r} & 0 \\ 0 & T \end{bmatrix}. \tag{16c}$$

Finally, the image $(W, Q, Z)$ of $(\mathcal{A}, \mathcal{B})$ under the map $\mathfrak{m}_\eta$ is given by

$$W = -\mathcal{A}_{23}^{-1}\mathcal{A}_{22}, \quad Q = \mathcal{A}_{23}^{-1}, \quad Z = -\mathcal{A}_{23}^{-1}\mathcal{B}_2. \tag{16d}$$

We next define the following matrices.

$$C_{rc} = -\mathcal{A}_{13}Q, \quad C_c = -(\mathcal{A}_{22} + \mathcal{A}_{23}\mathcal{A}_{33}Q), \tag{17a}$$
$$K_r = -\mathcal{A}_{11}, \quad K_{rc} = -(\mathcal{A}_{12} + \mathcal{A}_{13}W), \quad K_c = -(\mathcal{A}_{23}\mathcal{A}_{32} + \mathcal{A}_{23}\mathcal{A}_{33}W), \tag{17b}$$
$$E_r = \mathcal{A}_{13}Z + \mathcal{B}_1, \quad E_c = (\mathcal{A}_{23}\mathcal{A}_{33}Z + \mathcal{A}_{23}\mathcal{B}_3), \tag{17c}$$
$$V_c = \mathcal{B}_2. \tag{17d}$$

Finally, the image $(M, C, K, E, V) \in \mathcal{S}_{k_r+k_c}$ of $(\mathcal{A}, \mathcal{B})$ under the map $\mathfrak{m}_{lti}$ is given by

$$M = \begin{bmatrix} 0 & 0 \\ 0 & I_{k_c} \end{bmatrix}, C = \begin{bmatrix} I_{k_r} & C_{rc} \\ 0 & C_c \end{bmatrix}, K = \begin{bmatrix} K_r & K_{rc} \\ 0 & K_c \end{bmatrix}, E = \begin{bmatrix} E_r \\ E_c \end{bmatrix}, V = \begin{bmatrix} 0 \\ V_c \end{bmatrix}. \tag{17e}$$

B.2 MAPPINGS $\mathfrak{n}_{dynn}^{(l)}$ AND $[\mathfrak{n}_{dynn}^{(l)}]^{-1}$:

We next describe the bijective mapping

$$\mathfrak{n}_{dynn}^{(l)} : \left( \mathcal{M}^{(l)}, \mathcal{C}^{(l)}, \mathcal{K}^{(l)}, \mathcal{W}^{(l)} \right) \mapsto \left( M^{(l)}, C^{(l)}, K^{(l)}, E^{(l)}, V^{(l)} \right). \tag{18a}$$

First note that $\mathcal{W}^{(l)}$ is composed of $w_i^{(l)}$ (and similarly $\mathcal{M}^{(l)}, \mathcal{C}^{(l)}, \mathcal{K}^{(l)}$). Next, we partition $w_i^{(l)}$ as

$$w_i^{(l)} = \left[ \ e_i^{(l)} \ \big| \ v_i^{(l)} \ \big| \ -k_{i,i+1}^{(l)} - c_{i,i+1}^{(l)} \ \big| \ \cdots \ \big| \ -k_{i,n_l-1}^{(l)} - c_{i,n_l-1}^{(l)} \ \right] \tag{18b}$$

to define $e_i^{(l)} \in \mathbb{R}^{1 \times d_i}$, $v_i^{(l)} \in \mathbb{R}^{1 \times d_i}$, $k_{i,j}^{(l)}, c_{i,j}^{(l)} \in \mathbb{R}$ for $i \in \{0, \cdots, n_l - 1\}$ and $j \in \{i+1, n_l - 1\}$. Additionally, let $k_{i,i}^{(l)} := k_i^{(l)}$, $c_{i,i}^{(l)} := c_i^{(l)}$ for all $i \in \{1, \cdots, n_l - 1\}$. Finally, define the image $\left( M^{(l)}, C^{(l)}, K^{(l)}, E^{(l)}, V^{(l)} \right)$ of $\left( \mathcal{M}^{(l)}, \mathcal{C}^{(l)}, \mathcal{K}^{(l)}, \mathcal{W}^{(l)} \right)$ under the map $\mathfrak{n}_{dynn}^{(l)}$ as

$$
\begin{aligned}
M^{(l)} &= \begin{bmatrix} m_0^{(l)} & & & \\ & m_1^{(l)} & & \\ & & \ddots & \\ & & & m_{n_l-1}^{(l)} \end{bmatrix}, \quad
C^{(l)} = \begin{bmatrix} c_{0,0}^{(l)} & c_{0,1}^{(l)} & \cdots & c_{0,n_l-1}^{(l)} \\ & c_{1,1}^{(l)} & & \\ & & \ddots & \vdots \\ & & & c_{n_l-1,n_l-1}^{(l)} \end{bmatrix}, \\[2ex]
K^{(l)} &= \begin{bmatrix} k_{0,0}^{(l)} & k_{0,1}^{(l)} & \cdots & k_{0,n_l-1}^{(l)} \\ & k_{1,1}^{(l)} & & \\ & & \ddots & \vdots \\ & & & k_{n_l-1,n_l-1}^{(l)} \end{bmatrix}, \quad
E^{(l)} = \begin{bmatrix} e_0^{(l)} \\ e_1^{(l)} \\ \vdots \\ e_{n_l-1}^{(l)} \end{bmatrix}, \quad
V^{(l)} = \begin{bmatrix} v_0^{(l)} \\ v_1^{(l)} \\ \vdots \\ v_{n_l-1}^{(l)} \end{bmatrix}.
\end{aligned} \tag{18c}
$$

For the inverse map $[\mathfrak{n}_{dynn}^{(l)}]^{-1}$, note that we can read off the elements $e_i^{(l)} \in \mathbb{R}^{1 \times d_i}$, $v_i^{(l)} \in \mathbb{R}^{1 \times d_i}$, $m_i^{(l)}, k_{i,j}^{(l)}, c_{i,j}^{(l)} \in \mathbb{R}$ for $i \in \{0, \cdots, n_l - 1\}$ and $j \in \{i, n_l - 1\}$ from given matrices $\left( M^{(l)}, C^{(l)}, K^{(l)}, E^{(l)}, V^{(l)} \right)$ as in equation equation 18c. The image $\left( \mathcal{M}^{(l)}, \mathcal{C}^{(l)}, \mathcal{K}^{(l)}, \mathcal{W}^{(l)} \right)$ of $\left( M^{(l)}, C^{(l)}, K^{(l)}, E^{(l)}, V^{(l)} \right)$ under the inverse map $[\mathfrak{n}_{dynn}^{(l)}]^{-1}$ is then given by setting $w_i^{(l)}$ as in equation equation 18b, $k_{i,i}^{(l)} := k_i^{(l)}$ and $c_{i,i}^{(l)} := c_i^{(l)}$.

B.3 MAPPING $\mathfrak{m}_h$:

We next describe the mapping

$$\mathfrak{m}_h : \mathcal{P}_{lti}^{state} \ni (A, B) \mapsto (\mathcal{M}, \mathcal{C}, \mathcal{K}, \mathcal{W}) \in \mathcal{P}_{dynn}^{hidden}. \tag{19a}$$

Since $(A, B) \in \mathcal{P}_{lti}^{state}$, $A$ is a block-diagonal matrix which is partitioned together with the appropriate partitioning of $B$ as

$$
A = \begin{bmatrix} A^{(1)} & & & \\ & A^{(2)} & & \\ & & \ddots & \\ & & & A^{(L)} \end{bmatrix}, \quad
B = \begin{bmatrix} B^{(1)} \\ B^{(2)} \\ \vdots \\ B^{(L)} \end{bmatrix}, \tag{19b}
$$

where $A^{(l)} \in \mathbb{R}^{d_l \times d_l}$, $B^{(l)} \in \mathbb{R}^{d_l \times d_i}$. For $l \in \{1, 2, \ldots, L\}$, we construct tuples $(\mathcal{M}^{(l)}, \mathcal{C}^{(l)}, \mathcal{K}^{(l)}, \mathcal{W}^{(l)})$ as

$$[\mathfrak{n}_{dynn}^{(l)}]^{-1} \circ \mathfrak{m}_{lti} : \left( A^{(l)}, B^{(l)} \right) \mapsto \left( \mathcal{M}^{(l)}, \mathcal{C}^{(l)}, \mathcal{K}^{(l)}, \mathcal{W}^{(l)} \right). \tag{19c}$$

The tuples $\left( \mathcal{M}^{(l)}, \mathcal{C}^{(l)}, \mathcal{K}^{(l)}, \mathcal{W}^{(l)} \right)$ define the image $(\mathcal{M}, \mathcal{C}, \mathcal{K}, \mathcal{W})$ of $(A, B)$ (see Definition 2) under the mapping $\mathfrak{m}_h$.

### B.4 MAPPING $\mathfrak{m}_o$:

We next describe the bijective mapping

$$\mathfrak{m}_o : (A, B, C, D) \mapsto (\Phi, \Psi) \in \mathcal{P}_{dynn}^{output}, \quad \text{where } (A, B) \in \mathcal{P}_{lti}^{state}, (C, D) \in \mathcal{P}_{lti}^{output}. \quad \text{(20a)}$$

We partition the block-diagonal matrix $A$ together with the appropriate partitioning of $B$ as done in equation equation 19b so that $A^{(l)} \in \mathbb{R}^{d_l \times d_l}, B^{(l)} \in \mathbb{R}^{d_l \times d_i}$. For $l \in \{1, 2, \dots, L\}$, we construct tuples $\left(W^{(l)}, Q^{(l)}, Z^{(l)}\right)$ via

$$\mathfrak{m}_\eta : (A^{(l)}, B^{(l)}) \mapsto (W^{(l)}, Q^{(l)}, Z^{(l)}). \quad \text{(20b)}$$

For $l \in \{1, 2, \dots, L\}$, and any positive integer $a$, we define the matrices:

$$P_\xi^{(l)} = \begin{bmatrix} I_{k_r^{(l)}} & 0 \\ 0 & I_{k_c^{(l)}} \otimes \begin{bmatrix} 1 \\ 0 \end{bmatrix} \end{bmatrix}, P_\eta^{(l)} = \begin{bmatrix} 0 \\ I_{k_c^{(l)}} \otimes \begin{bmatrix} 1 \\ 0 \end{bmatrix} \end{bmatrix}, \quad \text{(20c)}$$

$$T_a^{(l)} = \begin{bmatrix} I_a \otimes \begin{bmatrix} 1 & 0 \end{bmatrix} \\ I_a \otimes \begin{bmatrix} 0 & 1 \end{bmatrix} \end{bmatrix} \quad \text{(20d)}$$

We then construct the matrices $\mathcal{F}, \mathcal{Z}$ as

$$\mathcal{F}^{(l)} = \left[ \left( P_\xi^{(l)} + P_\eta^{(l)} \begin{bmatrix} 0 & W^{(l)} \end{bmatrix} \right) \quad \left( P_\eta^{(l)} \begin{bmatrix} 0 & Q^{(l)} \end{bmatrix} \right) \right] T_{n_l}^{(l)}, \quad \text{(20e)}$$

$$\mathcal{F} = \begin{bmatrix} \mathcal{F}^{(1)} & & \\ & \ddots & \\ & & \mathcal{F}^{(L)} \end{bmatrix}, \quad \mathcal{Z} = \begin{bmatrix} Z^{(1)}(t) \\ \vdots \\ Z^{(L)}(t) \end{bmatrix}. \quad \text{(20f)}$$

Finally, for $l \in \{1, 2, \dots, L\}$ and $i \in \{0, 1, \dots, n_l - 1\}$, construct $\phi_i^{(l)} \in \mathbb{R}^{d_o \times 2}$ as

$$\begin{bmatrix} \phi_0^{(1)} & \cdots & \phi_{n_1-1}^{(1)} \mid \phi_0^{(2)} & \cdots & \phi_{n_2-1}^{(2)} \mid \cdots \mid \phi_0^{(L)} & \cdots & \phi_{n_L-1}^{(L)} \end{bmatrix} = C\mathcal{F}, \quad \text{(20g)}$$

$$\Psi = C\mathcal{Z} + D, \quad \text{(20h)}$$

which complete the description of the image $(\Phi, \Psi)$ of $(A, B, C, D)$ under the mapping $\mathfrak{m}_o$.

## C  NUMERICAL EXAMPLE

### C.1  CONVECTION-DIFFUSION EQUATION

**Problem setup:**  The spatial domain is $[0, 10] \times [0, 9.5]$ with 20 grid points in each dimension. The right and left boundaries are periodic. The boundary conditions at the top and bottom boundaries are Dirichlet with $T(x, 0) = T(x, 9.5) = 0$. The initial condition $T(x, y, 0) = 0$. The velocities in x and y dimensions are given by $v_x = 0.6$ and $v_y = 0$, and the diffusivity $\mathcal{D} = 1.4$. Let $t_d$ be a uniform grid in $[0, 10]$ in steps of $0.1$. Heat is injected into the system via the source term which is obtained by interpolating the function: $100 \exp\left(-0.8\left((x - l/2)^2 + (y - l/2)^2\right)\right)\delta(t - 0.2)$, where $\delta$ is the discrete-time unit impulse. We choose a uniform grid in space, and the gradient and Laplacian operators are discretized with second-order finite differences at each grid point $(i, j)$ as $\frac{\partial T}{\partial x}\big|_{i,j} \approx \frac{T_{i+1,j} - T_{i-1,j}}{2h}$, $\frac{\partial T}{\partial y}\big|_{i,j} \approx \frac{T_{i,j+1} - T_{i,j-1}}{2h}$ and $\left(\frac{\partial^2 T}{\partial x^2} + \frac{\partial^2 T}{\partial y^2}\right)\big|_{i,j} \approx \frac{T_{i+1,j} + T_{i-1,j} + T_{i,j+1} + T_{i,j-1} - 4T_{i,j}}{h^2}$. The spatially discretized form of equation equation 6 is an LTI system, where the state variable is represented by $T$. The spatial discretization scheme dictates the sparsity pattern and the elements of the state matrix $A \in \mathbb{R}^{400 \times 400}$. The other state-space matrices are $B = \mathcal{I}_{400}$, $C = \mathcal{I}_{400}$ and $D = 0$.

**Additional plots:**  Figure C.1 shows how the eigenvalues are clustered. Figure C.2 shows the Number of Function Evaluations (NFE) in the ODE solver averaged over the neurons of each horizontal layer. As the state dynamics are decoupled across the horizontal layers, the horizontal layers that require a lower NFE to solve the ODE to a prescribed tolerance are not forced to use a higher NFE as required by the other horizontal layers.

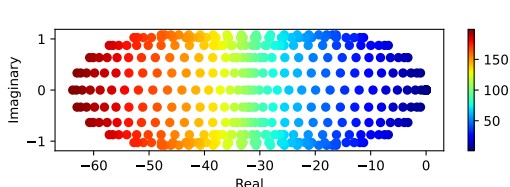

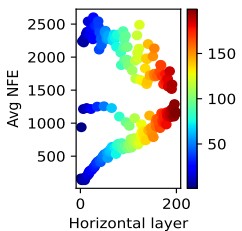

**Figure C.1:** Eigenvalue clusters of the state matrix (color bar indicates eigenvalue clusters)

**Figure C.2:** Number of function (right-hand side of the ODE) evaluations required by the ODE solver in each neuron averaged over horizontal layers (color bar indicates horizontal layers)

