# OpenReview forum: "Continuous-time neural networks for modeling linear dynamical systems"
_ICLR.cc/2024/Workshop/AI4DiffEqtnsInSci — AI4DiffEqtnsInSci @ ICLR 2024 Poster_

### Official Review · Reviewer_U715 · 2024-02-27
**Review for CONTINUOUS-TIME NEURAL NETWORKS FOR MODELING LINEAR DYNAMICAL SYSTEMS**

**Rating:** 6
**Confidence:** 2

**Review:**

In this work, an approach for obtaining a sparse neural network for an LTI system is introduced.
The short “workshop” version of this paper as submitted appears to be a high-level summary of a lengthier manuscript, with a large number of references to sections in the Appendix where the details and theory describing the approach is contained. Therefore, I cannot claim to fully understand the approach.
Nevertheless, I do think the contained ideas are interesting. I appreciate the author’s position that finding+fitting a neural network to an LTI system is an interesting exercise in obtaining sparse and accurate neural networks in a toy setting. However, one concern about abandoning a data-driven approach in favor of a (more principled) gradient-free approach is that this may scale quite poorly to more complex settings.
Comparing this approach to Neural Architecture Search baselines would be helpful as well, particularly any which attempt to enforce sparsity.

---

### Official Review · Reviewer_qt41 · 2024-02-27
**Are learning-free parametrized maps with neurons that are ODEs still neural networks?**

**Rating:** 6
**Confidence:** 3

**Review:**

The authors use so called dynamical neural network that are analytically fit to a LTI system. It thereby lifts a restriction of previous work that did not allow for clustered eigenvalues in the LTI system. Each eigenvalue block thereby maps, effectively, to an independent neural network.

The authors clearly state that they consider the present work as a stepping stone for more complex predictions. As such I can accept that no neural network training is involved in their work. Nonetheless, it poses for me the fundamental question why the maps considered in the paper with "neurons" that are ODES are neural networks and not just nonlinear maps that correspond to a LTI.

Minor: based on the usage in the text, most citations should be (X, 20XX) and not X (20XX

---

### Meta-Review · Area_Chair_N4CX · 2024-03-01

**Recommendation:** Accept (Poster)

**Metareview:**

Authors propose a solution to model complex dynamical systems using dynamical neural networks to linear time-invariant (LTI) systems. I recommend authors address the following for the camera-ready version: better justify the neural network framing, discuss scalability beyond LTI systems, expand details on the architecture, and potentially compare with neural architecture search methods.

---

### Decision · Program_Chairs · 2024-03-02

Accept (Poster)